# Association of Day-to-Day Variations in Physical Activity with Postprandial Appetite Regulation in Lean Young Males

**DOI:** 10.3390/nu11102267

**Published:** 2019-09-20

**Authors:** Takafumi Ando, Mitsuru Higuchi, Shigeho Tanaka

**Affiliations:** 1Department of Nutrition and Metabolism, National Institute of Health and Nutrition, National Institutes of Biomedical Innovation, Health and Nutrition, Tokyo 162-8636, Japan; 2Graduate School of Sport Sciences, Waseda University, Tokorozawa, Saitama 359-1192, Japan; 3Faculty of Sport Sciences, Waseda University, Tokorozawa, Saitama 359-1192, Japan; mhiguchi@waseda.jp

**Keywords:** physical activity level, day-to-day variation, accelerometer, preload test, ad libitum intake, energy sensing, postprandial satiety, energy turnover

## Abstract

Previous studies have shown that habitual physical activity improves postprandial appetite regulation. We evaluated the direct association between physical activity level (PAL) and postprandial appetite regulation, and the effect of day-to-day variations in PAL on improving postprandial appetite regulation in lean young males. Fourteen young male adults wore a triaxial accelerometer for at least 6 consecutive days to evaluate their PAL. Two random liquid preload tests were performed on separate days to evaluate the competence of postprandial appetite regulation. In the preload test, participants ate sandwiches ad libitum 75 min after drinking one of two liquids containing different energy densities. When a participant had an adequate regulation of their postprandial appetite, the difference in energy intake from sandwiches was expected to be close to the energy gap between both liquids. Average daily PAL (*r* = −0.558, *p* < 0.05), but not the SD of PAL, which is indicative of the day-to-day variations in PAL (*r* = −0.437, *p* > 0.1), correlated with the difference in energy intake from the sandwiches. In addition, higher average PAL was closer to the energy gap between the two liquids. These results suggest that average daily PAL, rather than day-to-day variations in PAL, predicts inter-individual variation in postprandial appetite regulation, at least for lean young males.

## 1. Introduction

Energy expenditure (EE) resulting from physical activity (PA) changes dynamically moment by moment. However, on a long-term basis, changes in body weight are not very large [1,2], which is partially due to the incessant regulation of energy homeostasis. Strong evidence suggests that regular high levels of PA or regular exercise indirectly influence energy homeostasis by altering appetite regulation [3], as previously indicated by Mayer [4]. Thus, PA may improve the accuracy of postprandial appetite regulation in response to calories from food. In other words, PA may improve the competence of energy sensing mechanisms [5,6,7,8]. Needless to say, the accuracy of energy homeostasis systems should influence weight change. Such physiological evidence strengthens the recommendation for regular PA for obesity prevention.

However, the above physiological studies were either exercise intervention studies [7,8] or studies in habitual exercisers or non-exercisers [5,6]. As such, the studies did not investigate the association between daily physical activity level (PAL) and postprandial appetite regulation, even though recent PA recommendations and strategies for obesity prevention have already shifted toward a daily basis of PA that includes lifestyle-related PA in addition to exercise. Moreover, the influence of day-to-day variations in PA on weight management remains unclear. While daily dynamic PA changes can directly influence body weight changes, to the best of our knowledge, no studies have demonstrated that large day-to-day variability in PA contributes to greater weight gain. This may suggest that individuals with larger day-to-day variability in PA exhibit a higher competence of postprandial appetite regulation than individuals with smaller day-to-day variability in PA. In fact, in the majority of individuals with high PAL, day-to-day variations in EE along with changes in PA may be greater than in individuals with low PAL.

Herein, we investigated the association of (1) PAL and (2) day-to-day variations in PAL with postprandial appetite regulation as evaluated by a single-blinded energy preload test and use of a triaxial accelerometer.

## 2. Materials and Methods

This study was conducted at the Graduate School of Sport Sciences, Waseda University and the National Institute of Health and Nutrition in Japan (NIHN) from 2009 to 2011. The study protocol was approved by the Ethical Committee of the NIHN (approval number: 20090831-01). Fifteen non-obese, young Japanese men were enrolled and provided written informed consent prior to participation in this study. We realize that the number of valid days of PA measurement in a subject did not meet our criteria once the subject was discharged from the study; therefore, we additionally recruited a new subject to meet the sample size (*N* = 14) upon subject discharge. Study participants were recruited by word of mouth, were non-shift workers, and had no chronic diseases affecting metabolism, such as diabetes, metabolic disease, or digestive disease.

### 2.1. Experimental Design

Figure 1a shows a schematic of the study protocol. At least 6 days before the preload test, anthropometric measurements, including bioelectrical impedance analysis, using Inbody720 (Biospace Japan, Tokyo, Japan) and physiological functional tests including a maximum oxygen uptake (VO_2_ peak) test by cycle ergometer, were performed after at least 2 h of fasting at the Graduate School of Sport Sciences, Waseda University. The participants then wore an accelerometer on their waist for evaluation of habitual PA until the end of the second preload test. Two preload tests were randomly performed, including a high-energy-dense liquid preload and a low-energy-dense liquid preload (Figure 1b). The random sequence for the preload tests was generated using Microsoft Excel. Three hours before energy preloading, all participants ate a provided meal (500 kcal; 15% of kcal from protein, 25% from fat, and 60% from carbohydrate) as breakfast at their homes, before attending study visits at the NIHN 1 h before energy preloading. On an energy preload test day, foods and drinks other than the provided breakfast and water were not permitted prior to arrival. At least 30 min before energy preloading, the participants were required to lie down (but not allowed to sleep) for approximately 25 min to avoid any effects of PA just prior to preloading and to unify the previous conditions. Two minutes before energy preloading, the participants provided baseline subjective appetite scores, and then again every 15 min. Participants were not allowed to eat or drink prior to eating the sandwiches after energy preloading. Seventy-five minutes after preloading, the participants consumed egg sandwiches until they felt full (ad libitum eating). Each visit was separated by at least 1 week.

### 2.2. Conditions and Dietary Treatments

Table 1 shows the contents of each preload liquid. The total energy contained in the high-energy-dense load and the low-energy-dense load liquids was 594 kcal and 321 kcal, respectively. The energy difference between these liquids was 273 kcal. The weight of both liquids was 450 g; the energy density was 1.32 kcal/g for the high preload and 0.713 kcal/g for the low preload. Macronutrient composition of the liquids was 8% of kcal (11.7 g) from protein, 48% (31.7 g) from fat, and 44% (65.6 g) from carbohydrates for the high preload and 16% (12.9 g) of kcal from protein, 47% (16.9 g) from fat, and 37% (29.4 g) from carbohydrate for the low preload; that is, the percentage of energy from fat and the amount of protein content were similar for both liquids. In addition, the smell, texture, and taste of both liquids were made to be as similar as possible. The liquids for preloading were made approximately 1 h before drinking and were cooled in the refrigerator at approximately 4 °C until 10 min before drinking. Participants were instructed to drink the liquids within 5 min at their own pace, but most participants drank the liquids within 2 min.

Egg sandwiches (Lunchpack, Yamazaki Baking Co., Ltd., Tokyo, Japan) were used for the ad libitum meal because the macronutrient composition (16% of kcal from protein, 30% from fat, and 54% from carbohydrate) of the sandwiches was similar to a general Japanese diet [9]. The energy content of each sandwich was either 115 or 135 kcal, due to a change in the product to a new package before all experiments had been fully completed. There were no intra-individual changes in the sandwich packages. The weight of each sandwich was measured every test day and the mean was observed to be approximately 56 g. We therefore used the mean sandwich weight (56 g) to calculate the weight of any leftover food. During the ad libitum eating period, each participant stayed alone and ate sandwiches in the cabin to isolate them from the other participants (except staff). At the first ad libitum eating, participants were provided 5 pieces of sandwich on the plate. Then, a sandwich was constantly added to the plate whenever they ate a piece of sandwich. Participants were permitted to eat sandwiches and drink water as much as they liked until they felt full. In addition, participants were allowed to eat sandwiches at their own pace. The weight of leftover sandwiches and water was measured in grams to calculate food and water consumption.

If a participant had a high competence of postprandial appetite regulation, the difference in energy intake from sandwiches was expected to be close to energy gap between both liquids.

### 2.3. Subjective Appetite 

A self-designed 11-point numerical rating scale (NRS) was used for assessing subjective appetite (e.g., fullness: 0 = no feelings of fullness, 10 = extreme feelings of fullness). The NRS included questions about gastric fullness (i.e., does your stomach feel full?) and overall fullness (i.e., do you feel full?).

Each participant provided their subjective appetite scores via the NRS prior to the preload tests and every 15 min until 120 min after the preload, except during sandwich eating (0, 15, 30, 45, 60, 75, and 120 min after the preload). Appetite trajectories were evaluated by adjusting for baseline (at 0 min) values.

### 2.4. Physical Activity Assessment

After providing informed consent, each participant was instructed to wear a triaxial accelerometer (Active Style Pro HJA-350IT; Omron Healthcare, Kyoto, Japan) on their waist during waking hours (except for water-based activities) for at least 6 consecutive days until the end of the study if possible. Participants were instructed to keep track of the time of attaching and detaching the device, getting into bed and waking up, and any intentional and intense activities, such as sports and exercise every day. Processing and calculation of PA data were carried out as previously reported [10]. The accelerometer recorded PA in metabolic equivalents (METs) for each 10 s time window (10 s epochs). The recorded PA data of ≥60 consecutive minutes of zero MET were defined as non-wear periods. The remaining periods with zero MET were assigned a value of 0.9 MET as basal metabolic rate. Participants with at least 6 valid days with ≥600 min of wear time per day were included in the following analysis. Finally, PAL was calculated using the sum of the MET values. Day-to-day variation in PAL was evaluated by SD of daily PAL and expressed as SD of PAL. The PA data were summarized as the accumulated time spent in each of the following categories: sedentary behavior (SB), ≤1.5 METs; light intensity PA (LPA), 1.6–2.9 METs; moderate intensity PA (MPA), 3.0–5.9 METs; and vigorous intensity PA (VPA), ≥6.0 METs. Daily PA data were obtained when participants performed VO_2_ peak tests and pre-load tests were excluded from the dataset.

### 2.5. VO_2_ Peak Assessment

VO_2_ peak was measured using a 1 min incremental exercise test on a cycle ergometer (Aerobike 75XL, Combi Wellness, Tokyo, Japan). Expired gas was sampled and measured by indirect calorimetry (Aeromonitor AE210, Minato Medical Science, Tokyo, Japan).

Before the test, participants performed a 5 min warm-up to reach a targeted heart rate (HR) of 110 beat/min, and the exercise load to achieve that HR was used as the load at the beginning of the test. Exercise load was increased by 15 W every minute until exhaustion. Considering proficiency in cycle exercise, cadence was chosen between 60 to 80 r/min based on each subject at warm-up. VO_2_ was considered “peak” if two of the following criteria were met: (1) measured HRmax ≥ age-predicted HRmax-10 beats/min; (2) VO_2_ increased by <100 mL/min during a 1 min increment; (3) RERmax was ≥1.10; and/or (4) Borg Scale maximum was ≥19. Finally, VO_2_ peak was expressed relative to body weight.

### 2.6. Statistical Analysis

The main analysis of this study was a correlation analysis between PA and the energy gap between ad libitum intake of sandwiches. We therefore based our sample size calculation on results from a previous preliminary study that examined the relationship between PAL and the differences in ad libitum intake in response to high- and low-energy preloads in 9 males [11]. We used an expected Pearson’s coefficient of 0.64 for the relationship between SD of PAL and the differences in ad libitum intake in response to high- and low-energy loads, with a power set at 80% and an α level (two-tailed) of 5%. This gave a sample size of 14 subjects.

A Pearson (partial) correlation analysis was performed to assess the associations among variables such as postprandial fullness, energy intake from sandwiches, PA variables, and VO_2_ peak. Also, a linear mixed effect model with the Bonferroni correction method was performed to assess the main effect of the differences in energy load and PAL or SD of PAL, and the interaction effect of these on postprandial fullness. A paired t-test was performed to assess the difference of appetite score after drinking between the two liquids. All analyses were performed by SPSS 25 (IBM SPSS, Armonk, NY, USA). Data are presented as mean ± SD.

## 3. Results

### 3.1. Subject Characteristics and Missing Values

Table 2 shows the subject characteristics. As the NRS for one subject could not be obtained, postprandial appetite was evaluated using 13 subjects.

### 3.2. Physical Activity and Sandwich Intake Results

Table 2 shows the results of PA variables and ad libitum sandwich intake. Mean wear day and wear time of the activity monitor were 14.2 ± 9.0 days and 903.3 ± 70.6 min, respectively. Mean PAL was 1.72 ± 0.15. Mean SD of PAL was 0.187 ± 0.099. A moderate association was shown between PAL and SD of PAL (*r* = 0.668, *p* < 0.01). In addition, PAL was slightly associated with SD of LPA (*r* = 0.526, *p* = 0.05) and SD of VPA (*r* = 0.649, *p* < 0.05), but not SD of MPA (*r* = 0.413, *p* > 0. 1) and SD of SB (*r* = 0.350, *p* > 0.2). SD of PAL was more strongly influenced by SD of VPA (*r* = 0.926, *p* < 0.0001) or SD of MPA (*r* = 0.834, *p* < 0.001) than by SD of LPA (*r* = 0.534, *p* < 0.05) or SD of SB (*r* = 0.443, *p* > 0.1). VO_2_ peak was more strongly associated with SD of PAL (*r* = 0.761, *p* < 0.01) compared with PAL (*r* = 0.519, *p* = 0.057). There was no association of PAL or SD of PAL with wear time or wear day (*p* > 0.2 for all). Energy intake from sandwiches was 102 kcal higher under the low preload condition than under the high preload condition. This mean difference was 170 kcal from the calculated energy gap between the two liquids.

### 3.3. Associations among Postprandial Fullness, Physical Activity, and the Competence of Postprandial Appetite Regulation

For the low load condition, there were significant correlations between PAL or SD of PAL and energy intake from sandwiches: for PAL, *r* = 0.568, *p* < 0.05; for SD of PAL, *r* = 0.754, *p* < 0.01. Meanwhile, for the high load condition, significant correlations were only observed between SD of PAL and energy intake from sandwiches: for PAL, *r* = 0.313, *p* > 0.2; for SD of PAL, *r* = 0.583, *p* < 0.05. Even after adjusting for fat-free mass, SD of PAL, but not PAL (*p* > 0.1 for both conditions), had correlations with energy intake from sandwiches under both conditions: low load, *r* = 0.772, *p* < 0.01; high load, *r* = 0.549, *p* = 0.052. In addition, there were no interactions between PAL or SD of PAL and either of the conditions. Also, PAL and SD of PAL exhibited the strongest correlation with energy intake from sandwiches for both conditions compared with the other activity variables (e.g., VPA) or SD of the other activity variables (e.g., SD of VPA). VO_2_ peak correlated with energy intake from sandwiches under both conditions: low load, *r* = 0.664, *p* < 0.01; high load, *r* = 0.582, *p* < 0.05.

Higher PAL (*r* = 0.653, *p* < 0.05, Figure 2a), but not SD of PAL (*r* = 0.321, *p* > 0.2, Figure 2b), was found to correlate with the difference in changes in gastric fullness at 75 min (from 0 min). PAL was also associated with the difference in energy intake from sandwiches (*r* = −0.557, *p* < 0.05), and was close to the value of the energy gap between the two liquids (Figure 2c). However, the difference in energy intake from sandwiches was not associated with SD of PAL (*r* = −0.449, *p* > 0.1, Figure 2d), VO_2_ peak (*r* = −0.245, *p* > 0.3), or any of the PA categories (e.g., MPA; *p* > 0.05 for all). When adjusted for PAL, neither the association of SD of PAL (*r* = −0.087, *p* > 0.7) nor VO_2_ peak (*r* = 0.061, *p* > 0.8) with energy intake from sandwiches was significant.

Changes in gastric fullness at 75 min (from 0 min) were slightly, but not significantly, correlated with energy intake from sandwiches under the low load condition (*r* = −0.477, *p* = 0.09; Figure 3a), but not under the high load condition (*r* = −0.281, *p* > 0.3; Figure 3a). Changes in overall fullness at 75 min (from 0 min) were slightly correlated with energy intake from sandwiches for both conditions: low load, *r* = −0.497, *p* = 0.08; high load, *r* = −0.691, *p* < 0.01; Figure 3b. In addition, the slopes of these associations were nearly the same for both conditions (Figure 3b). The difference in changes in gastric fullness at 75 min (from 0 min) predicted the difference in energy intake from sandwiches (*r* = −0.661, *p* < 0.05, Figure 3c), while the difference in changes in overall fullness was not associated with the difference in energy intake from sandwiches (*r* = −0.430, *p* > 0.1, Figure 3d).

### 3.4. Postprandial Fullness and Physical Activity Results

Figure 4a,c,e,g show the associations of PAL or SD of PAL with the trajectory of gastric or overall fullness after energy load under both conditions. Figure 4b,d,f,h show the associations of the difference of energy load and PAL or SD of PAL with the difference of gastric or overall fullness at 75 min after energy load and mean gastric or overall fullness during 15–75 min after energy load. When divided into two groups by the difference of PAL, the gastric fullness response to energy load was clearly different between the loads in individuals with higher PAL but not in those with lower PAL (Figure 4a). There were significant differences in gastric fullness at 75 min after energy load and mean gastric fullness during 15–75 min after energy load between energy loads only in those individuals with higher PAL (Figure 4b). A significant difference was also found for overall fullness at 75 min after energy load between energy loads only in those individuals with higher PAL (Figure 4d).

When divided into two groups according to the difference of SD of PAL, the gastric fullness response to energy load was also shown to be different between the loads in individuals with higher SD of PAL, but not in those with lower SD of PAL (Figure 4e). There were significant differences in gastric fullness at 75 min after energy load and mean gastric fullness during 15–75 min after energy load between the different energy loads only in the individuals with higher SD of PAL (Figure 4f). On the other hand, overall fullness after the energy load was shown to be lower in those individuals with higher SD of PAL, regardless of energy loads, compared with those with lower SD of PAL (Figure 4g). There was a significant main effect of SD of PAL on overall fullness during 15–75 min after energy load (Figure 4h).

## 4. Discussion

The present study sought to clarify the association of (1) daily PAL and (2) day-to-day variations in PAL assessed by an accelerometer with postprandial appetite regulation in lean young males. Our results show a clear correlation between PAL and the energy gap of the sandwiches between conditions, whereas day-to-day variations in PAL (SD of PAL) did not demonstrate a significant correlation with the energy gap of the sandwiches between conditions. Therefore, our results suggest that the competence of postprandial appetite regulation may be up-regulated by regular and moderately-high PAL but not by large daily PA changes, at least for lean young males. These results were not inconsistent with the results of previous studies that suggested that regular high levels of PA or regular exercise indirectly influences energy homeostasis by altering appetite regulation [4,5,6,7,8].

The plausible mechanisms underlying the association between “average” PAL and the competence of postprandial appetite regulation may be explained by postprandial gastric fullness, which may be related to gastric distention and gastric emptying. There were significant differences in gastric fullness at 75 min after energy load and mean gastric fullness at 15–75 min after energy load between energy loads only in those individuals with higher PAL, and PAL was associated with the difference in gastric fullness just prior to ad libitum intake between the two conditions. In addition, the difference in gastric fullness strongly predicted the difference in sandwich intake between the conditions. This association is likely mediated by the difference in postprandial gastric fullness that may be supported by the postprandial glucagon-like peptide-1 (GLP-1) response. A previous study showed slight improvement in GLP-1 response to meal intake after an exercise intervention [12]. Increase in GLP-1 is associated with delayed gastric emptying [13], which can lead to gastric fullness. In contrast, there was no correlation between SD of PAL and difference in gastric fullness just prior to ad libitum intake. Therefore, larger day-to-day variations in PAL may not contribute to improvement of postprandial appetite regulation, at least via gastric fullness.

On the other hand, interestingly, based on the results of postprandial overall fullness (Figure 4h,g), larger day-to-day variations in PAL might contribute to a lack of postprandial overall fullness. Our results suggest that postprandial overall fullness after the preload in individuals with higher SD of PAL was significantly lower than that in individuals with smaller SD of PAL, and slightly lower than those with higher PAL, regardless of preload energy. Also, the associations of SD of PAL with energy intake from sandwiches in both conditions were stronger than that of PAL. These results may suggest that larger day-to-day variations in PAL resulted in more expectations of food intake prior to eating than smaller day-to-day variations in PAL or higher PAL. When expectation of food intake is beyond the requirement of energy intake, such an expectation is suggested to result from hedonic effects [14]. Because the hedonic system is constructed from the brain reward system, food expectation via hedonic effects should be obtained by past experiences. Even more interesting is the fact that energy intake from sandwiches was more strongly correlated with SD of PAL than PAL, regardless of energy load. These results suggest that larger day-to-day variations in PAL may more strongly influence the expectation of food intake prior to eating (e.g., via the hedonic system) than PAL. Two reasons should be assumed for this hedonic effect based on past experiences. First, individuals with larger day-to-day variations in PAL may be more exposed to negative energy balance when they exhibit higher PAL; consequently, their brains always seek to store more energy, and are therefore looking forward to obtaining more food. Second, individuals with larger day-to-day variations in PAL may be more familiar with eating larger portion sizes of food when they exhibit higher PAL as an attempt of mimicking higher PAL; consequently, their brains look forward to obtaining more food. These reasons could influence the hedonic system, which in turn modulates homeostasis and food expectations. These results highlight the significance in shifting interest from average PA to day-to-day variations of PA. Previous studies [3] (in addition to our study) have shown the superiority of higher PAL on the competence of energy density sensing delivered from meals. In contrast, however, a previous systematic review reported that high PA might not play a role in prevention of weight gain [15]. This inconsistency may be partially explained by the results of the current study. We found that a strong association between PAL and day-to-day variations in PAL is likely, so that the disadvantage of higher day-to-day variation on postprandial satiety may mask the benefit of higher PAL on postprandial satiety.

Another interesting physiological finding is that the difference in gastric fullness was well correlated with the difference in energy intake from sandwiches, whereas energy intake from sandwiches at each test was not sufficiently predicted by gastric fullness prior to eating, especially for the high energy density test, but was well predicted by overall fullness prior to eating. The reasons for this discrepancy remain unclear. However, based on our results, overall fullness prior to eating may determine overall food intake; that is, how much to eat. Gastric fullness prior to eating may determine the details of food intake; in other words, when to stop eating.

Our study has several limitations. First, subject characteristics were limited by the small sample size and lack of female participants. In addition, the reliability of the 11-point NRS used in this study was not evaluated. These limitations may lead to a high risk of false positives and negatives. Further, for clear evaluation of physiological responses, we selected individuals within a narrow range of age and body weight per the inclusion criterion of “young non-obese men.” Further studies utilizing larger sample sizes and a variety of ages, body compositions, and genders are needed. However, because the results of our study, such as the association between average PAL and postprandial appetite regulation, do not contradict the results of previous studies [5,6,7,8], we believe the findings of our study offer beneficial information for future studies and strategies for weight management. In addition, the duration of the PA assessment may need to be longer to evaluate the SD of PAL for several participants. Although no relationship was found between wear days of accelerometer and SD of PAL, there was a large difference in wear days between the longest (39 days) and shortest days (6 days). Because this was a cross-sectional study, causality could not be determined. Further epidemiological and/or interventional studies are needed to investigate the reproducibility of the results of the current study and to clarify the mechanisms and causality.

The present study offers originality in the assessment of PA, in which day-to-day variations in PA were evaluated using an accelerometer, and supports the conclusions of several studies [5,6,7,8], as well as a previous review [3], indicating an association between PAL assessed using an accelerometer and the competence of postprandial appetite regulation. Moreover, the evaluation of intensity and duration of PA using an accelerometer is one of the strengths of our study. Although the present study could not reveal clear evidence regarding the associations of PA sub-categories, such as sedentary behavior, with the competence of postprandial appetite regulation, these results should enhance the significance of the evidence regarding the degree of energy turnover for the alteration of the competence of postprandial appetite regulation.

## 5. Conclusions

Taken together, the results of the present study showed that higher average PAL, but not day-to-day variations in PAL, was associated with higher competence of postprandial appetite regulation in lean young males. The difference in postprandial gastric fullness may underlie these associations. In addition, larger day-to-day variations in PAL were associated with lower overall postprandial fullness, regardless of the energy density of food. This association may suggest the possibility that larger day-to-day variations in PAL may contribute to lack of postprandial satiety for lean young males.

## Figures and Tables

**Figure 1 nutrients-11-02267-f001:**
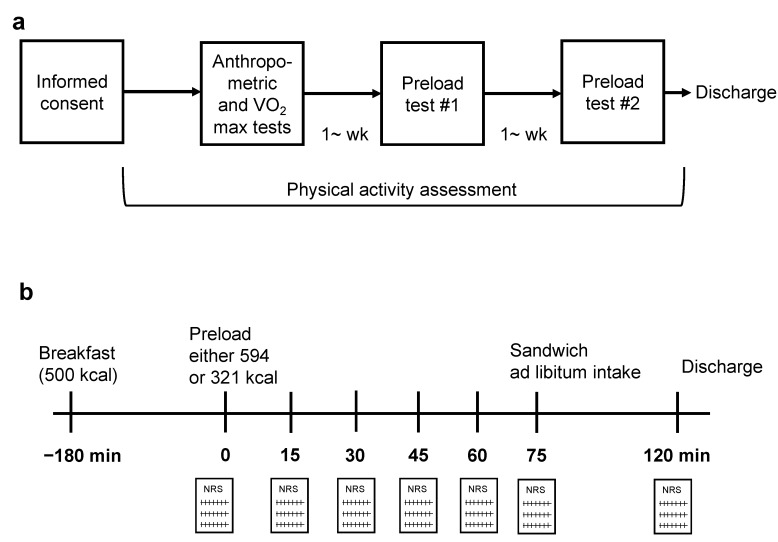
Schematic of the study protocol. (**a**) A flow diagram of the sequence of study procedures. Participants were randomly assigned two energy load conditions after performing anthropometric and VO_2_ max tests. (**b**) An overview of the preload experiment. Participants had a standardized breakfast 3 h prior to preload drinking. Seventy-five minutes after the preload, participants ate sandwiches ad libitum. Legend: NRS, numerical rating scale.

**Figure 2 nutrients-11-02267-f002:**
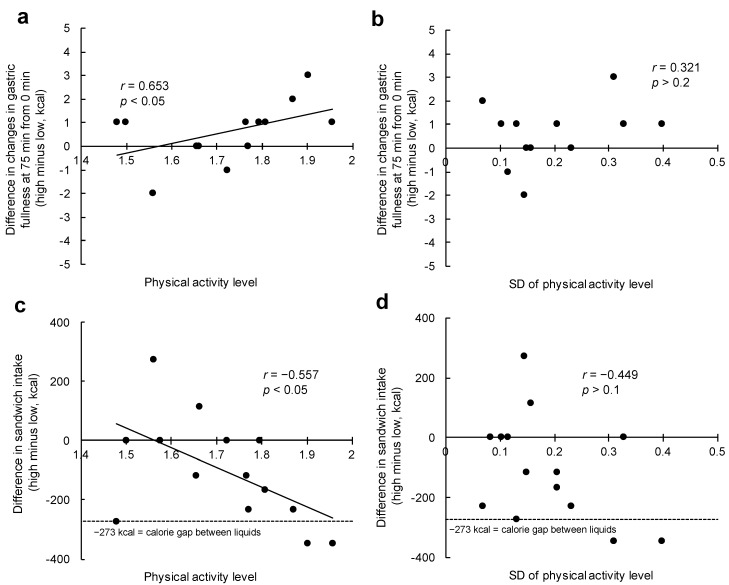
Association between physical activity and the difference in change in subjective appetite or energy intake from sandwiches between the two conditions. (**a**) Scatter plots between physical activity level and the difference in change in gastric fullness at 75 min (from 0 min). (**b**) Scatter plots between SD of physical activity level and the difference in change in gastric fullness at 75 min (from 0 min). (**c**) Scatter plots between physical activity level and the difference in the energy intake from sandwiches. (**d**) Scatter plots between SD of physical activity level and the difference in the energy intake from sandwiches.

**Figure 3 nutrients-11-02267-f003:**
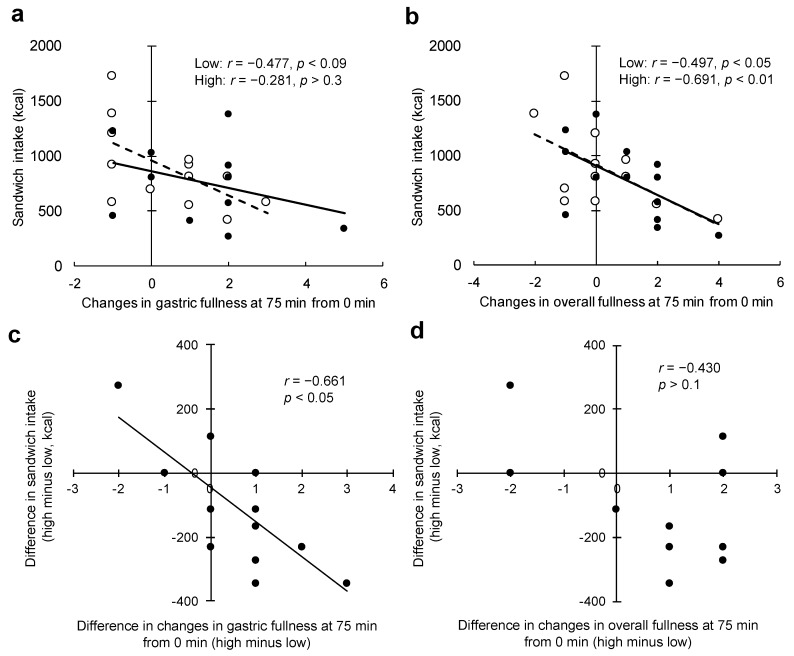
Association between subjective appetite and energy intake from sandwiches. (**a**) Scatter plots between changes in gastric fullness at 75 min (from 0 min) and energy intake from sandwiches. (**b**) Scatter plots between changes in overall fullness at 75 min (from 0 min) and energy intake from sandwiches. (**c**) Scatter plots between the difference in changes in gastric fullness at 75 min (from 0 min) and the difference in energy intake from sandwiches. (**d**) Scatter plots between the difference in changes in overall fullness at 75 min (from 0 min) and the difference in energy intake from sandwiches. Broken lines on (**a**) and (**b**) show the low energy load condition; dashed lines on (**a**) and (**b**) show the high energy load condition.

**Figure 4 nutrients-11-02267-f004:**
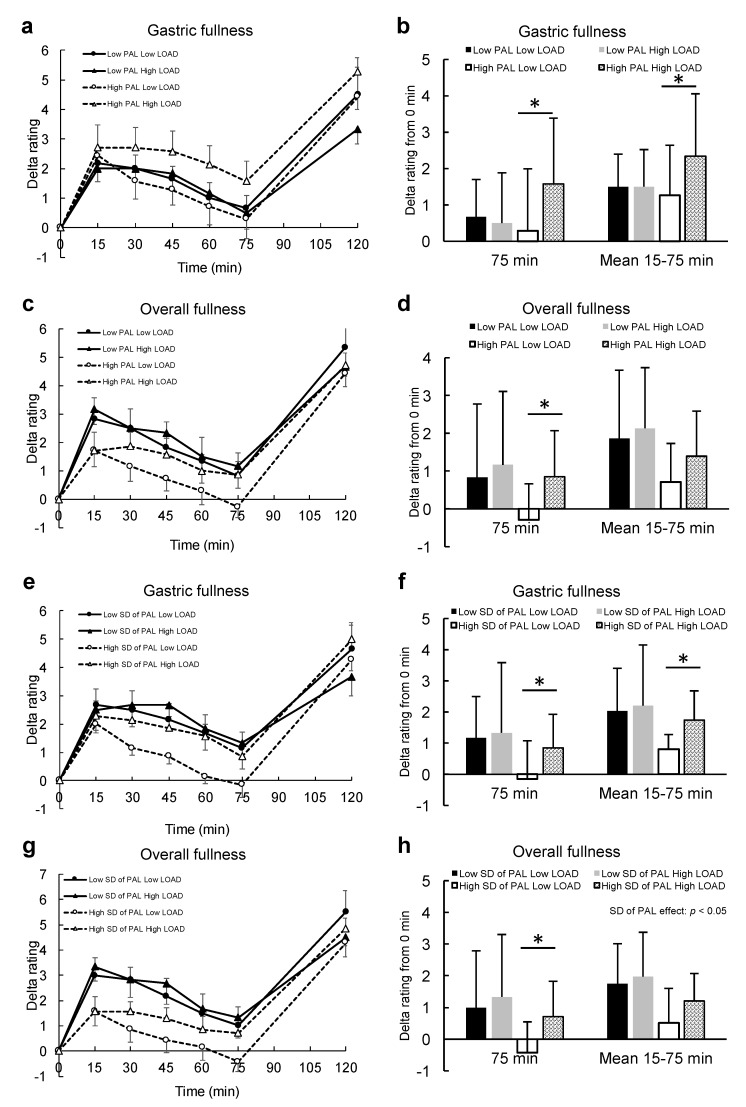
Results of various fullness variables after energy load. (**a**), (**c**), (**e**), (**g**) Trajectories of fullness variables following energy load in individuals with high or low PAL or SD of PAL. (**b**), (**d**), (**f**), (**h**) Mean values of the fullness variables at 75 min after energy load and during 15–75 min after energy load in individuals with high or low PAL or SD of PAL (mean ± SD). Legend: LOAD, energy load; PAL, physical activity level. * *p* < 0.05. Error bars for (**a**), (**c**), (**e**), and (**g**) show standard error. Error bars for (**b**), (**d**), (**f**), (**h**) show standard deviation.

**Table 1 nutrients-11-02267-t001:** Contents of liquids for each energy load test.

	Low Energy Liquid		High Energy Liquid	
	Contents	Weight (g)	Contents	Weight (g)
	Water	45	Water	30
	High fat milk	380	Low fat milk	310
	Maltodextrin	10	Fresh cream with a butterfat content of 45%	60
	Erythritol	15	Maltodextrin	35
	Vanilla extract	2 drops	Granulated sugar	15
			Vanilla extract	2 drops
Weight	450 g		450 g	
Energy	321 kcal		594 kcal	
Energy density	0.713 kcal/g		1.32 kcal/g	
Protein	12.9 g (16.1% of kcal)		11.7 (7.9% of kcal)	
Fat	16.9 g (47.3% of kcal)		31.7 (48.0% of kcal)	
Carbohydrate	29.4 g (36.6% of kcal)		65.6 (44.2% of kcal)	

**Table 2 nutrients-11-02267-t002:** Subject characteristics and results of physical activity and ad libitum intake.

Variables	Mean	±	SD	Range	
				Minimum	Maximum
Age (years)	23.6	±	3.2	20	29
Weight (kg)	62.8	±	5.8	54.9	72.8
Body mass index (kg/m^2^)	21.1	±	1.6	18.7	23.1
% fat mass (%)	13.9	±	4.7	6.7	22.4
VO_2peak_ (mL/min/kg)	45.3	±	8.3	29.7	57.3
Physical activity					
PAL	1.72	±	0.15	1.48	1.96
SD of PAL	0.187	±	0.099	0.069	0.398
Sedentary behavior (min/day)	584.0	±	105.5	439.5	803.5
LPA (min/day)	238.2	±	71.1	111.1	327.6
MPA (min/day)	78.5	±	23.5	52.6	121.1
VPA (min/day)	9.6	±	7.9	0.25	28.8
SD of sedentary behavior (min/day)	112.6	±	51.6	41.6	270.8
SD of LPA (min/day)	86.1	±	42.5	46.6	195.1
SD of MPA (min/day)	31.5	±	9.1	17.7	50.1
SD of VPA (min/day)	13.1	±	10.8	0.5	41.8
Wear time (min/day)	903.3	±	70.6	788.7	1013.7
Wear day (days)	14.2	±	9.0	6	39
Ad libitum intake					
Sandwich intake after high energy load (kcal)	770	±	334	274	1380
Sandwich intake after low energy load (kcal)	872	±	359	411	1725
∆ Sandwich intake (high–low, kcal)	−102	±	179	−345	274

Legend: LPA, light intensity physical activity; MPA, moderate intensity physical activity; PAL, physical activity level; VPA, vigorous intensity physical activity.

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
