# Peer review of "Association of Day-to-Day Variations in Physical Activity with Postprandial Appetite Regulation in Lean Young Males"

_nutrients, 2019, doi:10.3390/nu11102267_

Round 1
Reviewer 1 Report
Dear authors,
Many thanks for your submission to Nutrients. To my eyes this study is a small sample, non-preregistered, exploratory, randomised (although no method provided) crossover trial. I found some the text confusing, and I think overall the manuscript could be written more clearly with terms clearly defined when first used. Given that energy density sensing is in the title, I felt it was strange that this was not defined, and I am still not quite sure what it means and how you have operationalised it for measurement. I’ve provided more detailed comments below.
Abstract sentence one: consider correcting to: Habitual physical activity increases competence of energy density sensing which affects postprandial satiety.
Please define clearly what is meant by “competence of energy density sensing”
Please elaborate on this sentence:
“If that is indeed the case, the sensing of energy balance in individuals with high variability in PA should always be “active” to compensate for the residual energy balance.”
What does it mean to be “… “active” to compensate for the residual energy balance”?
Please provide a rationale for why “energy density sensing” is important?
This study was not preregistered so there is no way to establish whether the hypotheses were posed after knowledge of results. The study, in this case, must be view as exploratory rather than confirmatory.
Methods:
Please describe the trial design.
Were the participants students at the University the experiment was conducted? How were they chosen, by convenience or random sampling? Were there specific eligibility criteria that participants needed to have met? Please include these details in the manuscript.
How was the randomisation of participants to conditions performed? What method was used? Who generated the random allocation sequence, who enrolled participants, and who assigned participants to interventions? Were the researchers blinded to the random allocation of participants? Were outcome assessors blinded to the condition allocation of participants? Please include these details in the manuscript.
Pg. 2 line 75. Please correct: “Participants were not allowed to each or drink”.
“In addition, the smell, texture, and taste of both liquids were made to be as similar as possible.”
Was this tested? For example, were the participants asked whether they detected any differences between the liquids?
Please provide validity and reliability information on the 11-point scale used to assess subjective appetite? Is this a self-designed measure?
In methods please briefly provide details of processing and calculation of PA data.
Please reconsider your used of “VO2 peak”, see: https://www.physiology.org/doi/full/10.1152/japplphysiol.01063.2016
Also reconsider the criteria used for a max effort: https://www.ncbi.nlm.nih.gov/pubmed/18027991
Please provide details of your sample size calculations. The sample size calculations must be based on the actual analysis performed. Therefore, the sample size calculation must be performed for a linear mixed effects analysis (for a within-subject, time x condition interaction). Also, data from only 9 males are used to inform the estimated effects. Please use larger sample studies or better still meta-analyses for a more realistic effect estimate. Also, sample size calculations gave a sample of 14, yet 15 were recruited to the study, why was this? Please explain this in methods.
Results:
Please provide units of measurement for mean PAL.
Please provide summary data for each timepoint, i.e. condition 1 vs. condition 2.
Were there any time effects, i.e. were there difference between week 1 and 2 regardless of the condition allocated to?
Please provide estimated effect sizes and 95% confidence intervals for the assessment of the main effect of the differences in energy load and PAL or SD of PAL, and the interaction effect of these on postprandial fullness.
In figure 2, please provide alternatives to bar plots. Show your data points. For example, use violin, sina, or beeswarm plots with jittered data points.
In figure 3, please provide the key, black dots vs. white dots, solid line vs. dashed line.
Pg. 9 line 52. “Changes in gastric fullness at 75 min (from 0 min) was slightly correlated to energy intake from 252 sandwiches under the low load condition (r = −0.477, p = 0.09”. You have set your alpha at 0.05, so this is not slightly correlated. It is a non-significant correlation, by your own criterion.
How was multiple comparisons adjust for? There are so many analyses in the results section and given that the study is almost certainly under powered (80% power is even a pretty low bar), there is high risk of false positives and negatives.
I’d like the results to be better structured. In the statistical analysis section of results lay out the order of and details about comparisons and analyses that were performed. For consistency this should match the hypotheses made in the introduction. The results should then follow this structure.
Discussion:
Considering the exploratory design (i.e., not preregistered), low sample size, and the many analyses presented, it is difficult to interpret the results of this experiment.
You have described the study as cross-sectional, from your methods I would describe this study as a randomised crossover study.
Do you think not having a control condition, where participants are not given any drink, is a limitation?
Describe how a small sample may have influenced your results.
Author Response
Response to Reviewer #1
We appreciate the reviewer’s constructive comments and suggestions for improving the manuscript. We considered all of the reviewer’s comments carefully and revised the manuscript accordingly. Revisions and additions per the reviewer’s comments are denoted in red font in the revised manuscript, while our responses to the reviewer’s comments are provided below.
Comment 1 (for overall): Many thanks for your submission to Nutrients. To my eyes this study is a small sample, non-preregistered, exploratory, randomised (although no method provided) crossover trial. I found some the text confusing, and I think overall the manuscript could be written more clearly with terms clearly defined when first used. Given that energy density sensing is in the title, I felt it was strange that this was not defined, and I am still not quite sure what it means and how you have operationalised it for measurement. I’ve provided more detailed comments below.
Response:
We thank the reviewer for the insightful comments and suggestions. First, we believe our study can be regarded as an observational study because our aim was to investigate the association between physical activity and postprandial appetite regulation, although our protocol included a randomized crossover test to assess the inter-individual variability in postprandial appetite regulation. In addition, prior to 2015 in Japan, there was not aneed to register our study with any Clinical Trials Registrydue to the previous Ethical Guidelines for Epidemiological Research and Ethical Guidelines for Clinical Research, issued by the Ministry of Education, Culture, Sports, Science and Technology and the Ministry of Health, Labour and Welfare in Japan. As such, our study was not registered. Also, we feel the wording “energy density sensing” could easily lead to confusion and misunderstanding among the readers of our manuscript; therefore, we changed this wording to “postprandial appetite regulation” throughout the manuscript. Furthermore, we have re-written the Abstract, Introduction, and Discussion sections to improve readability and to revise the aim to be more explorative, per the reviewer’s comments below.
Comment 2 (for abstract): Consider correcting to: Habitual physical activity increases competence of energy density sensing which affects postprandial satiety.
Please define clearly what is meant by “competence of energy density sensing”
Response:
We thank the reviewer for the comment. To avoid confusion, we revised the wording “competence of energy density sensing” to “postprandial appetite regulation” and added a sentence to the Methods section of the Abstract to explain what we mean by “postprandial appetite regulation.”
Comment 3 (for introduction):Please elaborate on this sentence: “If that is indeed the case, the sensing of energy balance in individuals with high variability in PA should always be “active” to compensate for the residual energy balance.”What does it mean to be “... “active” to compensate for the residual energy balance”?
Comment 4 (for introduction):Please provide a rationale for why “energy density sensing” is important?
Comment 5 (for introduction):This study was not preregistered so there is no way to establish whether the hypotheses were posed after knowledge of results. The study, in this case, must be view as exploratory rather than confirmatory.
Response: Per the reviewer’s comment,we revised the Introduction thoroughly to improve readability and to avoid confirmatory descriptions.
Comment 5 (for Methods):Please describe the trial design.
Response:The trial design is described in Figure 1. We are happy to provide additional information or details that the reviewer feels may be lacking.
Comment 6 (for Methods):Were the participants students at the University the experiment was conducted? How were they chosen, by convenience or random sampling? Were there specific eligibility criteria that participants needed to have met? Please include these details in the manuscript.
Response: Thirteen of the 15 study participants were students from several different universities (we are unsure of the exact number of distinct affiliations because we did not record the student’s affiliations, only their occupations [i.e., student, office worker, etc.]; however, the number of distinct affiliations is likely greater than 3). A few of the study participants may have been students at the university. The participants were recruited regardless of classes at the university that they were engaged in and their participation in the study did not affect their academic performance at the university. The study participants were non-shift workers and had no chronic diseases affecting metabolism, such as diabetes, metabolic disease, and digestive disease, as described in the Methods section of the manuscript. In addition, subjects were recruited by word of mouth. We have included this information in the Methods section as follows: “The study participants were recruited by word of mouth, were non-shift workers and had no chronic diseases affecting metabolism, such as diabetes, metabolic disease, and digestive disease.”
Comment 7 (for Methods):How was the randomisation of participants to conditions performed? What method was used? Who generated the random allocation sequence, who enrolled participants, and who assigned participants to interventions? Were the researchers blinded to the random allocation of participants? Were outcome assessors blinded to the condition allocation of participants? Please include these details in the manuscript.
Response: As we have indicated in the Introduction, and also per our previous comment, this study was not an interventional study, although we did adopt a single blinded test for the preload tests. The random sequence for the preload tests was generated using Microsoft Excel by the first author (T.A.). We have included this information in the Methods section.
Comment 8 (for Methods):Pg. 2 line 75. Please correct: “Participants were not allowed to each or drink”.
Response:We thank the reviewer for pointing out this typographical error. We have revised the sentence as follows: “Participants were not allowed to eat or drink prior to eating the sandwiches after energy preloading.”
Comment 9 (for Methods):“In addition, the smell, texture, and taste of both liquids were made to be as similar as possible.” Was this tested? For example, were the participants asked whether they detected any differences between the liquids?
Response:It took a month to prepare the two liquids until the research staff was unable to detect a definite difference in the smell, texture, and taste of the two liquids, and until it was clear that the 10 members of the research staff did not get ill (i.e., vomiting and diarrhea) after drinking either of the liquids. In addition, we did ask whether the participants could detect any differences during each test. In most cases, the study participants answered “I don’t remember the previous drink” or “I didn’t realize a difference.” Unfortunately, we did not record these comments, which is why they are not described in the manuscript.
Comment 10 (for Methods): Please provide validity and reliability information on the 11-point scale used to assess subjective appetite? Is this a self-designed measure?
Response: Yes, this 11-point scale was a self-designed measurement. Unfortunately, we did not conduct any study on thevalidity or reliability of the scale, although the scale was partially based on reports in the literature (IJO 2000, https://www.ncbi.nlm.nih.gov/pubmed/10702749 ). We have added “self-designed” to the Methods section describing the 11-point scale.
Comment 11 (for Methods):
In methods please briefly provide details of processing and calculation of PA data.
Response: Details of the processing and calculation of PA data are already included in the Methods section. Please refer to several sentences following “Processing and calculation of PA data were carried out as previously reported [10],” which include the majority of the details regarding the procedure.
Comment 12 (for Methods): Please reconsider your used of “VO2 peak”, see: https://www.physiology.org/doi/full/10.1152/japplphysiol.01063.2016. Also reconsider the criteria used for a max effort: https://www.ncbi.nlm.nih.gov/pubmed/18027991.
Response: We understand that there are various issues associated with measuring and evaluating VO2max and that these have been discussed in previous publications, as the reviewer has indicated. For the most part, we agree with the argument put forth in the article by Poole and Jones (JAP 2017). Nearly half of subjects did not reach VO2max (i.e., VO2did not reach the plateau) in our study because our subjects encompassed a wide range of physical activity levels to avoid an abnormal distribution and to investigate the association of physical activity and food intake; therefore, we used secondary criteria to evaluate and ensure VO2during the maximum effort of cycling, as discussed by Poole and Jones. We believe the phrase “VO2peak is no longer acceptable” from Poole and Jones’s article essentially means to “stop using VO2peak with nodefinite secondary criteria” to distinguish subjects who stopped due to lack of motivation or for some other unexpected reasons. This is why we stated our criteria as we did. Also, we understand several secondary criteria for assessing VO2peak are still being used; therefore, we believe our method for measuring VO2peak is adequate for evaluating subject’s exercise capability and performance.
Comment 13 (for Methods):
Please provide details of your sample size calculations. The sample size calculations must be based on the actual analysis performed. Therefore, the sample size calculation must be performed for a linear mixed effects analysis (for a within-subject, time x condition interaction). Also, data from only 9 males are used to inform the estimated effects. Please use larger sample studies or better still meta-analyses for a more realistic effect estimate. Also, sample size calculations gave a sample of 14, yet 15 were recruited to the study, why was this? Please explain this in methods.
Response: The main analysis of our study was a simple correlation analysis between physical activity and the energy gap between sandwiches, and was the only analysis considered when determining sample size for the study. We have included this explanation in the Statistical Analyses section and revised the Methods section accordingly.Also, when we originally performed this study (~10 years ago), there was no relevant study that was able to calculate the sample size for our analyses regarding correlations between physical activity and energy gap of ad libitumintake of sandwiches, which is why we used our previous small study to calculate the sample size. Moreover, we also realized that the number of valid days of physical activity measurement in a subject did not meet our criteria after the subject was discharged; therefore, we additionally recruited a new subject to meet the sample size upon subject discharge. We have included this information in the first paragraph of the Methods section. We also removed the sentence beginning with “one subject was excluded from…” from the Results section.
Comment 14 (for Results): Please provide units of measurement for mean PAL.
Response:There is no specific unit for PAL.
Comment 15 (for Results): Please provide summary data for each timepoint, i.e. condition 1 vs. condition 2. Were there any time effects, i.e. were there difference between week 1 and 2 regardless of the condition allocated to?
Response:We thank the reviewer for the insightful comments. However, we do not have the data requested by the reviewer because we did not record this data electronically. Thus, we cannot perform the analyses suggested by the reviewer. However, our study was not an interventional study; therefore, we believe the order of the conditions did not influence the energy gap of the sandwiches.
Comment16 (for Results): Please provide estimated effect sizes and 95% confidence intervals for the assessment of the main effect of the differences in energy load and PAL or SD of PAL, and the interaction effect of these on postprandial fullness.
Response:We understand the reviewer’s suggestion. However, the main analysis of our study was a simple correlation analysis between physical activity and the energy gap between sandwiches, and the amount of results may be almost too much, as suggested by the reviewer. Therefore, we have not added these results to the manuscript. In addition, we however revised and deleted the results of the relevant analyses because we found some miscalculations in the dataset.
Comment17 (for Results): In figure 2 (now Figure 4), please provide alternatives to bar plots. Show your data points. For example, use violin, sina, or beeswarm plots with jittered data points.
Response: We understand the reviewer’s concerns. However, the bar plots seem sufficient and are simple and easy to understand for the present purpose.
Comment 18 (for Results): In figure 3, please provide the key, black dots vs. white dots, solid line vs. dashed line.
Response:Per the reviewer’s request, we removed Figure 3 from the revised manuscript.
Comment 18 (for Results): Pg. 9 line 52. “Changes in gastric fullness at 75 min (from 0 min) was slightly correlated to energy intake from 252 sandwiches under the low load condition (r = −0.477, p = 0.09”. You have set your alpha at 0.05, so this is not slightly correlated. It is a non-significant correlation, by your own criterion.
Response:Per the reviewer’s comment, the p-value did not suggest statistical significance, but the effect size did show a correlation; therefore, we described these results as being “slightly” correlated, and although we believe this description to be adequate, we revised the text to read “slightly but not significantly correlated.”
Comment 19 (for Results): How was multiple comparisons adjust for? There are so many analyses in the results section and given that the study is almost certainly under powered (80% power is even a pretty low bar), there is high risk of false positives and negatives.
Response: We agree with the reviewer’s concern. Unfortunately, both during and before the experiment (~10 years ago), we did not consider the risk of false positives and negatives by multiple comparisons. Therefore, we have added this concern as a limitation of the study as follows: “This limitation might lead to a high risk of false positives and negatives.”
Comment 20 (for Results): I’d like the results to be better structured. In the statistical analysis section of results lay out the order of and details about comparisons and analyses that were performed. For consistency this should match the hypotheses made in the introduction. The results should then follow this structure.
Response:We revised the order of the Statistical Analyses and the Results section to improve readability.
Comment 21 (for Discussion): Considering the exploratory design (i.e., not preregistered), low sample size, and the many analyses presented, it is difficult to interpret the results of this experiment.
Response:We agree with the reviewer’s concerns. In addition to the description on the study limitation noted above, we also added the following sentence: “the main purpose of our study was the correlation analysis” into the Methods sections.
Comment 22 (for Discussion): You have described the study as cross-sectional, from your methods I would describe this study as a randomised crossover study. Do you think not having a control condition, where participants are not given any drink, is a limitation?
Response:Again, we believe our study should be considered as a cross-sectional study. We understand the reviewer’s suggestion that conditions not allowing for drinking may be worthwhile to obtain a better physiological understanding. However, our design exhibits a sort of control condition (so-called active control condition), that is, a low energy density preload condition; therefore, we believe our protocol is sufficient to meet minimum requirements.
Comment 23 (for Discussion): Describe how a small sample may have influenced your results.
Response: We added details on the risk of false positives and negatives per our previous response the reviewer’s Comment 19.

Reviewer 2 Report
The title of your paper is long and complicated. I suggest simplifying.
Abstract, lines 18-19: “Two random liquid preload tests were performed twice on separate days to evaluate competence of energy density sensing delivered from meals.” If the two preload tests were performed twice, it sounds like four preload tests were performed. In the methods description, it appears that a total of two preload tests were performed; therefore, I think you need to revise this sentence.
Line 23: I think you need to add something after the statement “with a value closer to the energy gap between the liquids” (i.e. either “with higher” or “with lower physical activity levels”)
Lines 33-35: “Strong evidence suggests that regular high levels of PA or regular exercise indirectly influences energy homeostasis by altering appetite regulation [3], especially competence of energy density sensing delivered from meals affecting postprandial satiety” – Here it would be helpful to state the direction by which PA influences appetite regulation, competency of energy density, etc. (i.e. does PA improve appetite regulation, does PA increase or decrease competence of energy density sensing?
Lines 43-47: This hypothesis statement is very long. Please try to shorten to simplify the hypothesis statement for the reader.
Somewhere in the introduction it would be of benefit to define what is meant by competence of energy density sensing.
Lines 54-55: “All methods were performed in accordance with relevant guidelines and regulations”. This statement is quite vague. I suggest either outlining the specific guidelines and regulations or deleting this sentence.
Line 63: subscript the “2” in “VO2peak”
Line 75: “Participants were not allowed to each or drink prior to eating the sandwiches as part of the study.” – This sentence needs revision.
Lines 91-92: “…the percentage of energy from fat and protein content were similar for both liquids” – Table 1, the % of energy from protein for the low-energy liquid is 16.1% and for the high-energy liquid is 7.9%. This appears to be quite different.
Line 112: “If a participant had a high competence of energy density sending delivered from meals,…” – Is “sending” the correct word you intended to use here?
Line 117: The questions “how does your stomach feel full?” and “how do you feel full?” do not seem to be translated properly into English. It would probably make more sense to delete the word “how” from each question.
Line 134: “according to a number of valid day of weekdays and weekends” – This needs re-wording
Lines 139, 140, 141, 147, 149,150: The “2” in “VO2” needs to be subscripted. Please make sure this is corrected throughout the remainder of the manuscript.
Line 149: “…VO2 increased by < 100 mL/min during a trial”. I think you mean “…during an 1-min increment” rather than “during a trial”.
Line 150: Change “Scalemax” to “Scale maximum”
Lines 174-176: The abbreviations “LPA”, “MPA”, “VPA”, and “SB” need to be defined. What do these abbreviations stand for and what physical activity cut-points (i.e. METS) were used for each?
Line 186: Change to “Results for postprandial fullness and physical activity”
Figure 3: Which condition is represented by the solid dots and which is represented by the open dots? Why are some solid dots large and some small?
Figures 3 A and B: In each figure, there are 14 black dots, but only 10 white dots. If there were 14 participants in the study, shouldn’t there be 14 dots for each condition?
Is Figure 3 even necessary? It doesn’t seem surprising that someone with higher physical activity level would consume more of the sandwiches. I think the correlations could just be presented in the text.
Line 226: Technically, a p-value of 0.052 is not statistically significant if p=0.05 is your alpha level.
Line 279: The sentence starting with “Unfortunately, although these results are inconsistent with our prime hypothesis…” needs to be re-written for clarity and grammar correction.
Lines 299-300: “Interestingly, based on the results of postprandial overall fullness, larger day-to-day variations in PAL might contribute to lack of postprandial overall fullness and consequently, overeating.” I am not sure if this statement is supported by your results. Figure 2 shows that individuals with high SD of PAL were able to detect gastric fullness difference between the two conditions compared to those with low SD of physical activity. Doesn’t this contradict your statement here?
Author Response
Response to Reviewer #2
We appreciate the reviewer’s constructive comments and suggestions for improving the manuscript. We considered all of the reviewer’s comments carefully and revised the manuscript accordingly. Revisions and additions per the reviewer’s comments are denoted in red font in the revised manuscript, while our responses to the reviewer’s comments are provided below.
Comment 1 (for title): The title of your paper is long and complicated. I suggest simplifying.
Response: Per the reviewer’s suggestion, we changed the title as follows: “Association of day-to-day variations in physical activity with postprandial appetite regulation in non-obese young males.”
Comment 2 (for abstract):
Abstract, lines 18-19: “Two random liquid preload tests were performed twice on separate days to evaluate competence of energy density sensing delivered from meals.” If the two preload tests were performed twice, it sounds like four preload tests were performed. In the methods description, it appears that a total of two preload tests were performed; therefore, I think you need to revise this sentence.
Response: We thank the reviewer for this comment. Yes, the reviewer is correct, and we have now removed the word “twice” from this sentence in the revised manuscript.
Comment 3 (for abstract): Line 23: I think you need to add something after the statement “with a value closer to the energy gap between the liquids” (i.e. either “with higher” or “with lower physical activity levels”)
Response:We thank the reviewer for pointing this out. The reviewer is correct, and we have now replaced “between the liquids” with “between the 2 liquids” in the revised manuscript to better clarify our intended meaning.
Comment 4 (for introduction):Lines 33-35: “Strong evidence suggests that regular high levels of PA or regular exercise indirectly influences energy homeostasis by altering appetite regulation [3], especially competence of energy density sensing delivered from meals affecting postprandial satiety” – Here it would be helpful to state the direction by which PA influences appetite regulation, competency of energy density, etc. (i.e. does PA improve appetite regulation, does PA increase or decrease competence of energy density sensing?
Comment 5 (for introduction):Lines 43-47: This hypothesis statement is very long. Please try to shorten to simplify the hypothesis statement for the reader.
Somewhere in the introduction it would be of benefit to define what is meant by competence of energy density sensing.
Response:We thank the reviewer for these comments, and have revised the Introduction section thoroughly in the revised manuscript.
Comment 6 (for Methods): Lines 54-55: “All methods were performed in accordance with relevant guidelines and regulations”. This statement is quite vague. I suggest either outlining the specific guidelines and regulations or deleting this sentence.
Response: Per the reviewer’s comment, we have deleted this sentence.
Comment 7 (for Methods): Line 63: subscript the “2” in “VO2peak”.Lines 139, 140, 141, 147, 149,150: The “2” in “VO2” needs to be subscripted. Please make sure this is corrected throughout the remainder of the manuscript.
Response: We have revised VO2peak throughout the manuscript.
Comment8 (for Methods): Line 75: “Participants were not allowed to each or drink prior to eating the sandwiches as part of the study.” – This sentence needs revision.
Response:We thank the reviewer for this comment. We have revised the sentence as follows: “Participants were not allowed to eat or drink prior to eating the sandwiches after energy preloading”
Comment 9 (for Methods): Lines 91-92: “...the percentage of energy from fat and protein content were similar for both liquids” – Table 1, the % of energy from protein for the low- energy liquid is 16.1% and for the high-energy liquid is 7.9%. This appears to be quite different.
Response: We apologize for this oversight. We meant “the percentage of energy from fat and the amount of protein content were similar for both liquids.” We have revised this as follows in the revised manuscript: “the percentage of energy from fat and the amount of protein content were similar for both liquids.”
Comment 10 (for Methods): Line 112: “If a participant had a high competence of energy density sending delivered from meals,...” – Is “sending” the correct word you intended to use here?
Response: We thank the reviewer for pointing out this oversight. Indeed, we meant “sensing” in this case, not “sending.” As we have changed the description from “energy density sensing” to “postprandial appetite regulation,” we therefore revised this wording as follows: “If a participant had a high competence of postprandial appetite regulation, the difference in energy intake from sandwiches should be close to the energy gap between both liquids.”
Comment 11 (for Methods): Line 117: The questions “how does your stomach feel full?” and “how do you feel full?” do not seem to be translated properly into English. It would probably make more sense to delete the word “how” from each question.
Response: We thank the reviewer for the recommendation, and have deleted the word per the reviewer’s suggestion.
Comment 12 (for Methods): Line 134: “according to a number of valid day of weekdays and weekends” – This needs re-wording
Response: We thank the reviewer for pointing this out and have deleted this wording in the revised manuscript.
Comment 13 (for Methods): Line 149: “...VO2 increased by < 100 mL/min during a trial”. I think you mean “...during an 1-min increment” rather than “during a trial”.
Response: We thank the reviewer for pointing this out, and have revised the phrase per the reviewer’s suggestion.
Comment14 (for Methods): Line 150: Change “Scalemax” to “Scale maximum”
Response: We thank the reviewer for pointing this out, and have revised the text accordingly, per the reviewer’s suggestion.
Comment 15 (for Results):Lines 174-176: The abbreviations “LPA”, “MPA”, “VPA”, and “SB” need to be defined. What do these abbreviations stand for and what physical activity cut- points (i.e. METS) were used for each?
Response: We have included these abbreviations in the Methods section (please see lines 265-268)
Comment 16 (for Results):Line 186: Change to “Results for postprandial fullness and physical activity”
Response: We thank the reviewer for the suggestion, and have revised the sub-title accordingly.
Comment 17 (for Results): Figure 3: Which condition is represented by the solid dots and which is represented by the open dots? Why are some solid dots large and some small? Figures 3 A and B: In each figure, there are 14 black dots, but only 10 white dots. If there were 14 participants in the study, shouldn’t there be 14 dots for each condition?
Is Figure 3 even necessary? It doesn’t seem surprising that someone with higher physical activity level would consume more of the sandwiches. I think the correlations could just be presented in the text.
Response:We apologize for not better describing the dots in the figure caption. Per the reviewer’s suggestion, we have deleted Figure 3 from the revised manuscript.
Comment17 (for Results): Line 226: Technically, a p-value of 0.052 is not statistically significant if p=0.05 is your alpha level.
Response: We have removed “significantly” from the sentence as follows:“Even when adjusted for FFM, SD of PAL, but not PAL (p > 0.1 for both conditions), had correlations with energy intake from sandwiches under both conditions: low load, r = 0.772, p < 0.01; high load, r = 0.549, p = 0.052).”
Comment 18 (for Discussions): Line 279: The sentence starting with “Unfortunately, although these results are inconsistent with our prime hypothesis...” needs to be re-written for clarity and grammar correction.
Response: We thank the reviewer for pointing this out. The reviewer is correct. Merging with comments from another reviewer, we revised the sentence as follows: “These results were not inconsistent with the results of previous studies indicating that regular high levels of PA or regular exercise indirectly influences energy homeostasis by altering appetite regulation [4-8].”
Comment 18 (for Discussions): Lines 299-300: “Interestingly, based on the results of postprandial overall fullness, larger day-to-day variations in PAL might contribute to lack of postprandial overall fullness and consequently, overeating.” I am not sure if this statement is supported by your results. Figure 2 shows that individuals with high SD of PAL were able to detect gastric fullness difference between the two conditions compared to those with low SD of physical activity. Doesn’t this contradict your statement here?
Response: We thank the reviewer for the insightful comment, and agree with the reviewer’s suggestion. We revised the descriptions in the Discussion and Conclusion sections to include less mention of the possibility of overeating.

Round 2
Reviewer 2 Report
Please note that the following line numbers refer to the marked-up revised manuscript:
Line 15: “…habitual that habitual…” Delete the first “habitual”
Figure 1b: Indicate in the legend what “NRS” stands for
Line 178: Change “an” to “a”
Line 182: Change to “a correlation analysis”
Line 208: Change this subtitle to “Results for physical activity and sandwich intake”
Line 342: “…whereas day-to-day variations in PAL (SD of PAL) did not…” There seems to be something missing from the end of this sentence (the sentence appears to be incomplete)
Author Response
Response to Reviewer #2
We appreciate the reviewer’s constructive comments and suggestions for improving the manuscript. We considered all of the reviewer’s comments carefully and revised the manuscript accordingly. Revisions and additions per the reviewer’s comments are denoted in red font in the revised manuscript, while our responses to the reviewer’s comments are provided below.
Comment 1 (for abstract): Line 15: “...habitual that habitual...” Delete the first “habitual”
Response: We thank the reviewer for pointing this out. We have deleted “habitual” from this sentence.
Comment 2 (for Figure 1b): Indicate in the legend what “NRS” stands for
Response: We thank the reviewer for this comment. We have added the description for NRS in the figure legend of Figure 1.
Comment 3 (for Methods): Line 178: Change “an” to “a”
Response:We thank the reviewer for pointing this out. We have revised this wording.
Comment 4 (for Methods):Line 182: Change to “a correlation analysis”
Response:We thank the reviewer for pointing this out. We have revised this wording.
Comment 5 (for Results):Line 208: Change this subtitle to “Results for physical activity and sandwich intake”
Response:We thank the reviewer for these comments. We have changed the subtitle per the reviewer’s suggestion.
Comment 6 (for Discussion): Line 342: “...whereas day-to-day variations in PAL (SD of PAL) did not...” There seems to be something missing from the end of this sentence (the sentence appears to be incomplete)
Response: We thank the reviewer for pointing this out. We have revised this sentence as follows:Our results show a clear correlation between PAL and the energy gap of sandwiches between conditions, whereas day-to-day variations in PAL (SD of PAL) did not have a significant correlation with the energy gap of sandwiches between conditions.”
